# Taiwanese Nuns and Education Issues in Contemporary Taiwan

**Yu-Chen Li**

The Graduate Institute of Religious Studies, National Cheng Chi University, Taipei 11605, Taiwan; dharma0958@gmail.com

**Abstract:** In this article, I discuss the Buddhist educational profile of nuns in contemporary Taiwan by introducing the development of monastic education for women. Taiwanese women's mass ordination created a Buddhist renaissance after postwar Taiwan, a national ordination system based on monastic discipline, as well as the revival of monastic education. Both ordination and monastic education are very strong institutional settings for women's monastic identity. Its findings, firstly, shed light on how the increased opportunities for women's education in Taiwanese Buddhism have continuously attracted young female university students. Secondly, these so-called scholarly nuns come to Buddhist academies as students and eventually become instructors. These scholarly nuns elevate the standards of their Buddhist academies and use their original academic specializations to expand the educational curriculum of their school. The role of scholarly nuns in contemporary Taiwan exemplifies that Buddhism provides educational resources for women, as educational resources enhance women's engagement in Buddhism.

**Keywords:** contemporary Taiwanese Buddhism; nuns; gender; monastic education; *bhikṣuṇī*s' identities

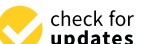



## 1. Introduction

In 1990 at Colombo University, a Theravāda monk said to his student, a Taiwanese nun named Xing-yan 性嚴, "You must be a very important figure or someone who will hold an important position once you return to Taiwan; otherwise how could you (a nun) study abroad?" She replied, "No, I'm just an ordinary Buddhist teacher. After I return to Taiwan, I won't change to any other position and will continue to teach at my original Buddhist academy. I do appreciate my study here for enriching my pedagogy so as to benefit my students in the future".[1]

The Theravāda monk's question reflects the idea that monastic leadership requires significant education and that most nuns are excluded from this path. In contrast, Taiwanese nuns enjoy advanced Buddhist education both as students and teachers.

In contemporary Taiwan, improvements in Buddhist nuns' education have promoted the profile of Buddhist nuns, increased their social support, and raised their self-esteem (Li 2016; Eichman 2011). The profile of postwar Taiwanese Buddhist nuns has also been the focus of increased scholarly attention. At least 30% received college education before the 1990s (Din 1996; Li 2005), and 70% owned graduate degrees in Taipei as of 2011, endorsing high esteem and social respect for Taiwanese nuns.[2]

Previously, much of the research regarding Taiwanese Buddhist academy education has focused on the history, curriculum, and academic goals (Huang 2008b; Borchert 2017). Moreover, studies tend to be more passive (if not male-centralized) analyses. From a gender studies perspective, I argue that research on Taiwanese monastic education should focus more on how female students, teachers, and faculties are involved in the educational system because the majority members of such Buddhist institutes are women.

## 2. Buddhist Education and Monastic Membership

Monastic education is designed for initiation into collective monastic life within Buddhist communities. Education, discipline, and initiation comprise the "Triple Platform

Ordination" (*santan dajie* 三壇大戒), specifically the initiation of *śrāmaṇera* and *śrāmaṇeri*, *bhikṣu* and *bhikṣuṇī*, as well as the receiver of *bodhisattva-śīla*.[3] For the monastic, and at every stage of membership from novice beginner to high-ranking monk, monastics undertake the specific responsibilities and education that correspond to their standing.

In practice, only those receiving full initiation are qualified to engage in certain facets of monastic education, and thus, much is withheld from *śrāmaṇeri* and laypeople. Without the appropriate knowledge, lower-level monastics cannot reprimand more eminent monastics and are disqualified from participating in monastic administration. In other words, the establishment of monastic education not only relates to an efficient ordination system but is also closely tied to monastic hierarchy in defining the obligations and rights of monastic members.

Historically speaking, monastic education has been almost always reserved for male members; thus, those religious women who were well-educated enjoyed privileged status. Contemporary scholars, such as Karma Lekshe Tsomo, who have been long devoted to promoting women's religious status, tend to emphasize nuns' education (Tsomo 1999). Moreover, the majority of research focusing exclusively on monastic education has taken the role of nuns as Buddhist teachers for granted and even assumed that education empowers women in the religious circle (LeVine and Gellner 2005). The scholarly emphasis on women's monastic education mainly recognizes its role in nuns' full monastic membership. I argue that the more understanding we have of Taiwanese nuns' engagement in education, the greater the chance for establishing a more modern, improved monastic education system for nuns.

Lack of education is usually a source of disadvantage for women in terms of social achievement; such is also the case in religious establishments. There are many cases in which religious orders excluded women from the clerical hierarchy for their illiteracy (Liu and Feng 2019). Ironically, though, religion itself very often offers women greater chances for education outside the public school system. For example, the reason why Buddhist education attracted many Taiwanese women in the 1950s to 1970s was its function as a public-school alternative. It was not until 1968 that the Taiwanese government prolonged compulsory education from six to nine years. Before this, Taiwanese parents with limited resources only invested in their sons' education, and so, the free tuition, room, and board offered by Buddhist institutions provided girls the chance for education.

These relatively small and often temporary Buddhist academies relied heavily on the support of their host monasteries. In return, the Buddhist academies regularly included monastic training in the curriculum and advocated the Dharma lineage inherited from the abbot or abbess of the host monastery. Gradually, regular Buddhist institutes, such as the Yuanguang Buddhist Academy (*Yuanguang foxueyuan* 圓光佛學院), the Fuyan Buddhist Academy (*Fuyan foxueyuan* 福嚴佛學院), the Fo Guang Shan Tsunglin University (*Foguangshan congling xueyuan* 佛光山叢林學院) (hereafter FGS),[4] the Chung-Hwa Institute of Buddhist Studies (*Zhonghua foyansuo* 中華佛研所), and the Lotus Buddhist Ashram and Institute of Sino-Buddhist Studies (*Lianhua xuefo yuan, Huafan foxue yanjiusuo* 蓮華學佛園，華梵佛學研究所), emerged (Chung-Hwa Institute of Buddhist Studies 2002; He 2006). These mainstream Buddhist institutes are open to both monastics and the laity, both male and female, except for the Lotus Buddhist Ashram Institute, which is only for women. Students learn advanced Buddhist training in a place financially and practically independent from the host monasteries. Instead of being part of a Dharma lineage (*famai* 法脈) in which Buddhist knowledge is transmitted from master to disciple, teachers and students become affiliated with each other within a lineage of Buddhist learning (*xuemai* 學脈).

## 3. Buddhist Academies and Nuns in Post-War Taiwan

In Buddhist traditions such as those found in contemporary Taiwan, where the monastic education system is institutionalized like a modern Western education system and where women are able to receive the same education as men, the issue of women's Buddhist education remains quite complicated. Since the conclusion of the Chinese Civil War (1927–1949),

Taiwanese Buddhist circles have emphasized monastic education as a continuation of the Buddhist reforms from the first half of twentieth century China, creating an important context regarding monastic education from this period (Jian 2001; Chan 2005).

Monastic education reform usually emerged as the first immediate strategy for Buddhist communities to resolve crises (Pittman 2001). As the social elite tried to confiscate monastic property to build modern public schools, since the end of the nineteenth century, one of the most influential Buddhist reformers, Master Taixu (1890–1947), urged Buddhist leaders to develop monastic education. He believed that Buddhism was in decline and attributed the crisis to the waning of monks' social status caused by lack of education. More specifically, Taixu urged his fellow Buddhists to devote themselves to more cultural, educational, and charity activities to save Buddhism (Lin 2001). Taixu's introduction of modern institutes of Buddhist learning were extremely important for Buddhist reform efforts. However, his plans only marginally included nuns' education. Though Taixu realized that traditional monasteries could not compete with the modern school system, he ignored the massive participation of women in Buddhist learning (Li 2020, p. 590).

Buddhist educational reforms in post-war Taiwan did not simply imitate those in mainland China but reflected aspects of orthodox Chinese Buddhism in pre-colonial Taiwan. Less than one hundred Chinese monks fled to Taiwan after 1949, but they maintained their dominance within the Buddhist Association of Republic of China (*Zhongguo Fojiao hui* 中國佛教會) (hereafter, BAROC), the only officially recognized national monastic representative organization. This led to a series of movements to purify the influence of Japanese Buddhism in order to restore the orthodox position of Chinese Buddhism (Jones 1999).

Most importantly, BAROC began to educating Taiwanese monks and nuns through the ordination system. The ordination system established and dominated by BAROC was a new invention gradually organized by Venerable Baisheng 白聖 (1904–1989) (Huang 2012). BAROC was a tentative aggregation of monks that had fled from China, including members of different areas and schools. BAROC leaders therefore decided to use canonical texts as the foundation to reestablish the ordination system, which allowed them to disregard problematic regional and sectarian differences. Before Martial Law ended in 1987, only BAROC could bestow ordinations and issue ordination certificates in Taiwan (Li (Forthcoming)).

From 1953 to 1987, the number of female ordinands far surpassed male, being four times more on average (Li 2008a, 2010). Some assistant nuns later became female monastic leaders famous for their oral interpretation of *Vinaya* scriptures during ordination. For example, Ven. Tianyi 天乙 (1924–1976) helped Baisheng in these training programs and was the first female ordination master in Taiwanese history. As a consequence, she became Baisheng's first Dharma heir (*fazi* 法子) (Jianye 1999; Li 2000, 2008b). More significantly, BAROC's reforms, which initially only addressed the full ordination of monks and nuns, caused a surge in popularity for Buddhist institutes and the study of canonical scripture. After the 1990s, most Taiwanese monasteries stopped running their own Buddhist academies. Large-scale Buddhist academies, such as the FGS Buddhist Academies, the Chung-Hwa Institute of Buddhist Studies, and the Lotus Buddhist Ashram and Institute of Sino-Buddhist Studies, took their place. These served as the basis for Buddhist universities that were founded later.

To sum up, in post-war Taiwan, Buddhist orders sought to ordain large numbers of women and established educational institutes to target young women and recruit new members (Huang 2008a; Li 2010). Taiwanese Buddhist institutes were usually open to both nuns and laywomen. Most institutes had one small class in each school-year until these students graduated and irregularly reopened the school for another group of students if they needed. In contrast to these small and short-lived institutions, several larger Buddhist orders also ran regular educational institutions, which transformed into Buddhist universities after 2000. Education, ordination, and Buddhist academies would shape the network for Buddhist nuns in Taiwan (Li 2016).

**4. From Scholarly Nuns (**學士尼**) to Religious Teachers (**宗教師**)**

In addition to Buddhist monastic education, which served the vehicle of ordination system, Taiwanese Buddhists also paid great attention to attracting young students to Buddhism. As Ven. Yinshun 印順 (1906–2005) pointed out, "To revive Chinese Buddhism, the focus should be placed on the youth, the intelligentsia, and the laity" (Yinshun 1970; Qiu 2000). This vision was reified in the establishment of Buddhist clubs in universities and winter and summer camps for college students. The first Buddhist student club was established under these conditions at National Taiwan University in 1960, and the number of such organizations increased to 73 by 1990 (Shengyan 1990).

Two Buddhist summer camps played pioneering roles in the 1960s and 1970s: that of the FGS order and the Learning Association of Vegetarian and Monastic Discipline (齋戒學會) (hereafter ZJXH) (Huang 2008a). Ven. Cihui 慈惠 and Ven Cijia 慈嘉, two nun disciples of Ven. Xingyun 星雲 who held master degrees from Japanese universities, became instructors at FGS Buddhist camps (Yongdong 2003; Li 2005). The founder of ZJXH, Ven. Chan-Yun 懺雲 (1915–2009), entrusted the Luminary Nunnery to manage female participants. Most of the women subsequently entered the Luminary Nunnery, making it the most famous nunnery of scholarly nuns (Din 1996; Huang 2008a).

The FGS's comprehensive program of Buddhist education aims to achieve their ideal of a Pure Land on earth. Although they attach great importance to classical Buddhist education, they reject sectarianism through their own concept of "establishing a common understanding across the Eight Buddhist schools" (八宗共榮) and put forth a particular style of monastic education, whereby they try to establish a common understanding across various sects (Manyi 2005). Furthermore, they offer students Japanese and English classes as they approach graduation in order to prepare them for an international career. Besides Buddhist academy education, FGS also consistently holds short-term training programs every year to help students refresh practical secular skills and reinvigorate their spiritual practice. The principles of humanistic Buddhism serve as FGS's blueprint for helping nuns develop in all aspects of cultural education and Dharma promotion (Chan 2005; Manyi 2005; Hui Kuan 2008).

The ZJXH summer camp strictly observes sexual segregation and requires all female students to leave Lianyin Temple 蓮因寺, where the camp is regularly held, by 4 p.m. The Lianyin Temple originally cooperates with the Luminary Nunnery (香光寺) and later also with the Yide Nunnery 義德寺 to accommodate female students attending the summer camps (Huang 2008a). Based on the convenient connection, two nunneries recruited many female college students from the summer camps and later developed into their own institutes. Campus Buddhist studies clubs and summer Buddhist camps have attracted so many female college students who then go on to receive tonsure that a special term has been created to refer to these Taiwanese nuns: scholarly nuns (學士尼, literally "nun with a bachelor's degree").[5]

The Luminary Institute aims at providing women Buddhist education and preparing them to serve as teachers in the Buddhist learning programs at the seven nationwide branches of the Luminary Nunnery (Jianye 2000).[6] In addition to traditional Buddhist monastic education, the nun students of the Luminary Institute also receive journalism, editing, and teaching training (Huang 2008b). The aim of the Buddhist education is to prepare students to attend the Luminary Institute and eventually move on to enter their own Buddhist orders (Huang 2008a, pp. 90–91). In a sense, the institute creates a new kind of modern Buddhist teacher rather than traditional masters preaching the Dharma and focusing on particular scriptures.

Income from ceremonial services is an important source of monastic economy, but the Chinese Buddhist modernization movement has viewed ceremonial services as an impediment to progress since the beginning of the 19th century. Those educated Taiwanese nuns receiving scripture-centered studies also follow the negative attitudes towards ceremonial services and ritual performs. As most monasteries and nunneries mainly relied on ceremonial services and are unable to provide academic work for their nuns, a tension between

immerged among monastic members. Those who are reluctant to perform ritual services tend to become "religious teachers", a position rarely seen at most small-scale nunneries.

For the disconnect between the sculptural-studies-centered education and monastic economy, graduating from Buddhist academy means certain kind of unemployment. The graduated nuns need to readjust to the ceremonial services. Even during their school years, these nun students are often accused as being not gregarious and lazy, leading their contemporary leave from their own nunneries to maintain the pace of student life. Therefore, some nuns tried to extend their student status as long as possible, such as keeping on transferring among different institutes to avoid the embarrassing situation after graduation, which created the term "professional students" to refer the phenomena.

Among Taiwanese Buddhist academies, the Luminary Nunnery established by scholarly nuns (more than 75%) also surpassed other Buddhist orders in Taiwan with the number of nuns having doctoral degree (at least 10 by 2000). These doctoral nuns are all scholarly nuns benefited by their bachelor degree in various professions. Their profession may be critical for spreading Dharma in modern society, but they are usually not included in the curriculum of the Buddhist Institute. After graduation, almost all of the nuns now holding these doctoral degrees chose to teach in universities, claiming that their positions there could be more influential than staying in Buddhist institutes. As these Ph.D. *bhikṣuṇī*s are overqualified for general monastic education, instead of dissociating with them, the Luminary Nunnery leaders gradually figured out a strategy to preserve good relationship with these high-profile nuns. The abbess Wuyin 悟因 strongly supported these nuns in their search for advanced study in various fields at colleges and universities across the world. These professional nuns develop their religious career outside monastic circles as university professors and research fellows, while the Luminary Nunnery continues to recognize them as members with a relatively flexible appointment. In this way, this group of nuns has adopted a new identity as religious teachers (宗教師).

Taiwanese nuns who have been educated at Buddhist academies take divergent paths in their practice. On the one hand, some turn towards the development of a modern, professionally certified Buddhism and create *bhikṣuṇī* groups with this goal in mind, such as the Luminary Nun's Organization, an organization concerned with promoting Buddhist education in society. On the other hand, some, such as those at the Nanlin Nun's Center南林尼僧苑), focus on strict observation of the precepts and ascetic practice in an effort to rectify the decline of *Vinaya* (monastic discipline 戒律). Their inward and outward approaches illustrate the two extremes of scholarly nun practice in Taiwan: one ascetic and inward focused and the other progressive and focused on society. On both sides, Buddhist nuns come with specializations in various fields from higher educational institutions and then supplement their study with Buddhist academy education. Interestingly, neither education level nor area of specialization seems to affect the number of nuns who favor ascetic or humanistic Buddhism.

## 5. Nun Teachers and Nuns' Education in Taiwan

The deeper tension between genders in the monastery can be largely attributed to the norms of Buddhist tradition, which promote male leadership in the order. However, the social changes in contemporary Taiwan have compromised male authority in the monastery, where nuns not only outnumber monks but have also surpassed them in terms of education level. The following case focused on Fuyan Academy (福嚴佛學院) demonstrates this reshaping of gender hierarchy and division of labor in Taiwan.

Ven. Yinshun 印順 (1906–2005) established the Fuyan Academy for monks in 1961; however, there were only two male novice students, and the classroom was filled with an audience of nuns. Therefore, the Fuyan Academy changed the system for nuns. In the following decades, Fuyan Academy trained many nun students and teachers in light of Yinshun's style of scriptural studies. Among these nuns, Zhaohui 昭慧 (b. 1957) has been famous for her interpretation of the work of Ven. Yinshun. She continues to be known as a

scholarly nun, having graduated from the Taiwanese Normal University, attended the FGS summer camp, received tonsure, and being devoted to teaching Yinshun's work.

When Ven. Zhenhua 真華 (1922–2021) was appointed as the sixth dean of Fuyan Academy in 1985, he decided to "rectify" the situation by limiting enrollment to only monks, a move that radically altered the situation. Nuns who were students and teachers had to move out and find new lodgings and affiliations. In response, Ven. Zhaohui gathered those nun teachers and graduated students dismissed from the Fuyan Academy and established a new institute for women, the Hongshi Academy (弘誓學院). Ven. Zhaohui, who was at that time serving as a professor in the Department of Religious Studies at Hsuan Chang University (玄奘大學), created a joint program for these two institutions. Afterwards, many of her nun students and colleagues attained master degrees in her department (Li 2008b).

In short, though higher levels of education bring Buddhist nuns new social roles, such as instructor in dedicated Buddhist education institutes, they are rarely accepted into the hierarchical power structures of monasteries that remain under the jurisdiction of Buddhist monks. The identity of Buddhist nuns in Taiwan tends to stress their roles as "religious teachers" after they leave the family and serve in other spheres. Their status as "religious teachers" also transitions their career from religious vocation to professional occupation.

After the 1990s, most Taiwanese monasteries stopped running their own Buddhist academies, and many large-scale Buddhist academies appeared on a regular basis. Since 2000, to promote Buddhist education, Taiwanese Buddhists have donated money and resources to establish six Buddhist colleges and universities, including the Huafan College of Technical Science (華梵理工學院), the Medical Colleges of Tsu Chi Ji University (慈濟醫學院), the Nanhua College of Humanities (南華人文學院), Hsuan Chuang University, Foguang University (佛光大學), and the Dharma Drum Institute of Liberal Arts (*Fagu wenli xueyuan* 法鼓文理學院). Many monks and nuns attend these Buddhist colleges and universities to pursue their Buddhist education rather than Buddhist institutes. Meanwhile, certain well-established Buddhist institutes continue to cooperate with these colleges to promote their education. Even though a greater emphasis is placed on academic achievement through this exchange, ironically, most Buddhist institutes insist more on their scriptural and philosophical approach.

Taiwanese nunneries have gradually created their own features for various religious goals, specifically for Buddhist orders with more than 200 female monastic members. They have tried to distinguish their own monastic education in specific ways. For example, the students and teachers at the Luminary Academy aim for Buddhist social education; the Ling Jiou Mountain Buddhist society (靈鷲山佛教基金會) established a museum of world religions to promote interreligious dialogues, and the Buddhist Institute of the Nanlin Nunnery (南林僧苑) offers more specific *Vinaya* education for nuns. Other relatively large Buddhist organizations, such as the FGS monastery and the Sangha University of the Dharma Drum order, gather data on students' post-graduation employment opportunities, which range from classics' translation, digital publication, meditation instructors, and monastic management, among others. These specialized Buddhist academies also advertise new kinds of jobs taken on by students. Most importantly, nuns make up 80% of the monastic order and contribute to more diverse and professional development of Buddhism based on Buddhist learning rather than sectarianism.

According to my previous research, in the 1990s, there were at least 26 Taiwanese *bhikṣuṇī*s who received scholarships from the Chinese Buddhist Association for Safeguarding the Sangha (中華佛教護僧協會) to study abroad (as compared to only four Taiwanese *bhikṣu* scholarship recipients); this number does not include nuns who were supported by other monasteries. It is quite common in Taiwan for Buddhist academies to send nuns to study abroad. Some monasteries send monks and nuns to study abroad in order to establish overseas outposts and start local branches of their school. Others allow nuns to use scholarships or family support in order to take leave and study abroad. These nuns gradually become critical personnel in the process of internationalization of Taiwanese Buddhism.

Most of the nuns with a Ph.D. continue to devote themselves to Buddhist education either at their original Buddhist academy or at a Buddhist university. Although Buddhist education in Taiwan was institutionalized in the 2000s, with the establishment of Buddhist universities, large Buddhist academies remain active even to the extent of cooperating with Buddhist universities. Nun students at Buddhist institutes often go on to graduate at Buddhist universities. Most monasteries require tonsured college graduates to go to Buddhist institutes for further education. After the 2000s, the Buddhist education system in Taiwan continued to produce more and more nun teachers who hold doctoral degrees from domestic Buddhist academies and universities. Given that domestic tuition fees are cheaper than those abroad, there are more domestically educated nuns than foreign-educated nuns in Taiwan. In addition, given the high number of graduates, there is fierce competition for employment at Buddhist academies and universities.

These so-called Buddhist universities are unique because they are funded by Buddhist organizations, but on the whole, they are average public universities. They only have a few departments related to Buddhism, so naturally, there is a limit on the number of staff and students in Buddhist studies. In order to meet the requirements of the Ministry of Education, which promotes appointments of staff with a degree gained abroad, educators with foreign degrees are favored, making it harder for nuns with domestic degrees to pursue an academic career. That is not to say that domestically educated nuns are refused teaching positions, only that it is harder for them to attain such positions. More specifically, a fault has emerged between faculty and resources.

Instead of severing ties with their nunneries, these doctoral nuns established their own small institutes and supported their lives by various teaching jobs. They often become part-time teachers at both Buddhist institutes and public schools. As the abbesses of their new nunneries, they enjoy more freedom to arrange their monastic lifestyle, such as to change the designation of their nunneries into "lecture halls" (講堂) to focus on Buddhist teaching program and publication. Like the scholarly nuns who are continuously affiliated with their masters' nunneries, these abbesses who have doctoral degrees and affiliate with other Buddhist academies run various Buddhist classes as well as publish their books and address their small lecture halls or Buddhist abodes as satellite instructors.

Let us now return to Ven. Xinyan of the anecdote that opens this paper. She first studied in Sri Lanka for a year and later obtained a Ph.D. in Buddhist Studies in the United Kingdom before returning to Taiwan and assuming the role of provost at her Buddhist institute, Yuanguang Buddhist Academy. She is not a case of a satellite instructor. She has been financially supported throughout by the academy, which seeks to install well-educated nuns in leading positions. Indeed, there have been three other nuns at the academy who have received doctorates abroad in Japan and the United States. All four nuns returned to teach at Yuanguang Academy with a clear mission: to develop the Buddhist educational content and programs at the academy. Xingyan credits the dean for letting her concentrate on her studies and obtain her degree as well as understanding that Taiwan's Buddhist education system is becoming increasingly academic, so there is a need to cultivate qualified teachers.

Today, most of the junior-high-level institutions of the above-mentioned Buddhist academies were converted into preparatory schools for foreign students, with courses offered for the students with respective native languages to learn classic Chinese and Mandarin. In this way, nun teachers and instructors at Buddhist institutes also stimulate the sense of gender equality for foreign monastic students.

## 6. Conclusions

There are few women in the global Buddhist community that have received a Buddhist education, and the level of education that they enjoy is generally not comparable to that of men. If monastic education is compulsory and becomes part of religious life, then it is an invaluable resource for nuns, yet few have been allowed to receive such an education. This is not the case in Taiwan. From doctrine to practice and from self-cultivation to the

cultivation of others, Taiwanese Buddhist nuns can acquire knowledge and skills through education and thus benefit themselves and others. Among the well-known Buddhist nun leaders of modern times, many have received a Buddhist education and are able to spread the Dharma and lead others.

At first, less than 100 Chinese monks came to Taiwan to preserve Chinese Buddhism after 1949. They then went on to reestablish the BAROC to form a new ordination system, opening the door for ordination and monastic education to Taiwanese women. This led to a large influx of women participating in Taiwanese Buddhism. Furthermore, this rare opportunity allowed for the gradual establishment of outstanding Buddhist nuns' groups, both in terms of quality and quantity.

Because of the increased opportunities for women's education, Taiwanese Buddhism continues to attract young female university students, the so-called scholarly nuns, who come to Buddhist academies as students and eventually become instructors. These scholarly nuns elevate the standards of their Buddhist academies and use their original academic specializations to expand the educational curriculum of their schools. That being said, Buddhist academies continue to prioritize a more traditional Buddhist education. Furthermore, the average Buddhist nunnery does not have enough suitable jobs for this supply of scholarly nuns and Ph.D. nuns. As a result, many have relied on their expertise and established separate, specialized Buddhist academies or entered the public education system to teach. The younger generation of Ph.D. nuns has established their own nunneries, where they serve as instructors and teach a Buddhist curriculum.

**Funding:** This research was funded by Taiwan Ministry of Science and Technology (MOST) on the research project of The Education System of FGS and Yuanguang Buddhist Academic in the Post-War Taiwan (MOST 109-2410_H-004-179).

**Acknowledgments:** I am indebted to Ven. Xing-yan and Miao-fan who kindly provided information in interviews. I would also like to thank Ester Bianchi and Nicola Schneider for their support, which helped produce and shape this article. However, all faults remain entirely my own.

**Conflicts of Interest:** The author declares no conflict of interest.

## Abbreviation

T　　*Taishō shinshū daizōkyō* 大正新修大經. *85 vols, Edited by Junjirō Takakusu* 高楠順次郎 *and Kaigyoku Watanabe* 渡邊海旭. *Tokyo: Taishō Issaikyō Kankōkai, 1924–1934.*

## Notes

[1]　I was told of this interaction during my fieldwork when I conducted interviews with the nun teachers at the Yuanguang Buddhist Academy (圓光佛學院) in Taoyuan in April 2022.

[2]　Lixiang Yao 姚麗香 conducted this investigation with the Taipei Branch of BAROC in November, 2011, preparing for the alms to *saṃgha* (monastic community) in the next 7th lunar month.

[3]　In the history of Imperial China, the Triple Platform Ordination system is attributed to Monk Daoxuan (道宣 596–667) who established the Ordination Plateform and the Tang court carried out in 765. Another resource points to the Song court, it is said, the whole system was nationalized in 1009. However, the official ordinaiton system has been supported by the government since the fifth century.

[4]　The FGS Tsunglin 'University' claims to follow the Buddhist discipline of traditional monasteries, but it was not yet recognized by the government as a university.

[5]　There was a scandal that happened in 1997, for 129 college girls received tonsure without their parents' agreement and hid at the Zhongtai Chan Monastery (Li 2005). Many Taiwanese criticized the monks of the Zhongtai Chan Monastery abduct these girls, as well as these hypnotized girls unfilial and superstitious. Indeed, the term scholarly nun first appeared in 1987, referring to those college students who received tonsure at the Fragrant Light Nunnery after attending the Buddhist summer camp held by ZJXH. Back to 1969, the FGS Summer camp already converted college girls to enter its order. The phenoma that massive tonsure of college girls at Buddhist summer camps is resulted from the increasing number of the Buddhist Fellowship on campous in the 1960s, strongly supported by the Buddhist circle for including the youth. The phenomenon sparked protests in 1997.

6    The Luminary Nunnery has established seven branches: Jiayi Huiguan 嘉義會館 in Jiayi City in 1981 (1984); Zichulin Convent 紫竹林精舍 in Kaohsiungin 1984 (1987); Anhui School 安慧學苑 in Jiayi County in 1988 (1988); Dinghui School 定慧學苑 in Miaoli County in 1995 (1995); Yinyi School 印儀學苑 in Taipei in 1997 (1997); Yanghui School 養慧學苑 at Taizhong City in 1998 (1998); and Xiangshan Nunnery 香光山寺 in Taoyuan City in 2003 (2003). The dates shown in the parentheses indicate the year their Buddhist courses began. The reason the majority are named as *xueyuan* 學苑 is to identify the locations as Buddhist educational institutions (Jian 1995).

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
