# Peer review of "Taiwanese Nuns and Education Issues in Contemporary Taiwan"

_religions, doi:10.3390/rel13090847_

Round 1

Reviewer 1 Report

Some remarks:

Note 3: the exact date of the introduction of the Triple Platform Ordination can be discussed, but it is certainly later than the fifth century.

Although the level of English seems quite fine to me, it might be good to have a final look by a native speaker (especially lines 206-211, 331-332 (unclear) and note 2).

Note 6 is unclear. It might be good to explain the event in more detail.

Line 192 speaks of nine branches, note 7 gives seven branches.

Some typos:

The word bhikṣuṇī is quite often misspelled as bhikṣunī (this might be a computer problem).

line 45: śrāmaṇeri should be śrāmaṇerī

line 139: FKS should be FGS (confer line 83)

line 140: Chunhua should probably be Chung-Hwa (confer line 83)

Author Response

Note 3: the exact date of the introduction of the Triple Platform Ordination can be discussed, but it is certainly later than the fifth century.

Clarified.

In the history of Imperial China, the Triple Platform Ordination system is attibuted to Monk Daoxuan (道宣596-667) who established the Ordination Plateform and the Tang court carried out in 765. Another resource points to the Song court, it is said, the whole system was nationalized in 1009. However, the official ordinaiton system has been supported by the government since the fifth century.

Although the level of English seems quite fine to me, it might be good to have a final look by a native speaker (especially lines 206-211, 331-332 (unclear) and note 2).

Note 6 is unclear. It might be good to explain the event in more detail.

Rerwrited.

Income from ceremonial services is an important source of monastic economy, but the Chinese Buddhist modernization movement has viewed ceremonial services as an impediment to progress since the beginning of the 19th century. Those educated Taiwanese nuns receiving scripture-centered studies also follow the negative attitudes towards ceremonial services and ritual performs. As most monasteries and nunneries mainly relied on ceremonial services and are unable to provide academic work for their nuns, a tension between immerged among monastic members. Those who are reluctant to perform ritual services tend to become “religious teachers,” a position rarely seen at most small-scale nunneries.

Line 192 speaks of nine branches, note 7 gives seven branches.

corrected according to suggestion.

Some typos:

The word bhikṣuṇī is quite often misspelled as bhikṣunī (this might be a computer problem).

I corrected the word  bhikṣuṇī.

line 45: śrāmaṇeri should be śrāmaṇerī

corrected.

line 139: FKS should be FGS (confer line 83)

corrected.

line 140: Chunhua should probably be Chung-Hwa (confer line 83)

corrected.

Reviewer 2 Report

The main question addressed is about the education of nuns in Taiwan, the various institutes where they receive education in Buddhism and the fact that many have degrees, including doctorates from regular secular universities and some from foreign universities.

I've not seen any discussion of this elsewhere. I was aware from my own field research on engaged Buddhism that in most of those groups nuns outnumber monks, in one of them by about 7:1, and Tzu-chi is all female though men contribute to running the organization. In only one group is there a more or less equal number of monks and nuns, and it is much smaller than the others.

Other comments:

129: Wasn't Zhengyan ordained by Yinshun before 1987?

166-168; Italics intended?

208-211; 214-216; awkward; appears that words have been left out.

Author Response

I've not seen any discussion of this elsewhere. I was aware from my own field research on engaged Buddhism that in most of those groups nuns outnumber monks, in one of them by about 7:1, and Tzu-chi is all female though men contribute to running the organization. In only one group is there a more or less equal number of monks and nuns, and it is much smaller than the others.

Other comments:

129: Wasn't Zhengyan ordained by Yinshun before 1987?

Zhengyan was self-tonsured, and was not allowed to join the Ordination ceremony. At the last moment, she met Yinshun and recieved his signiture as tonsure-master to become a legally recognized nun.

166-168; Italics intended?

Corrected according to suggestion.

208-211; 214-216; awkward; appears that words have been left out.

For the disconnect between the sculptural-studied centered education and monastic economy, graduating from Buddhist academy means certain kind of unemployment. The graduated nuns need to readjust to the ceremonial services. Even during their school years, these nun students are often accused as being not gregarious and lazy, leading their contemporary leave from their own nunneries to maintain the pace of student life. Therefore, some nuns tried to extend their student-status as long as possible, such as keeping on transferring among different institutes to avoid the embarrassing situation after graduation that create a term as “professional students” to refer the phenomena.
